# Synthesis of Analogs to A-Type Proanthocyanidin Natural Products with Enhanced Antimicrobial Properties against Foodborne Microorganisms

**DOI:** 10.3390/molecules28124844

**Published:** 2023-06-19

**Authors:** Antonio Cobo, Alfonso Alejo-Armijo, Daniel Cruz, Joaquín Altarejos, Sofía Salido, Elena Ortega-Morente

**Affiliations:** 1Department of Health Sciences, Faculty of Experimental Sciences, University of Jaén, Campus of International Excellence in Agri-Food (ceiA3), 23071 Jaén, Spain; acmolinos@ugr.es (A.C.); dcsaez@ujaen.es (D.C.); 2Department of Inorganic and Organic Chemistry, Faculty of Experimental Sciences, University of Jaén, Campus of International Excellence in Agri-Food (ceiA3), 23071 Jaén, Spain; aalejo@ujaen.es (A.A.-A.); jaltare@ujaen.es (J.A.)

**Keywords:** A-type proanthocyanidin analogs, flavylium chemistry, antimicrobial activity, antibiofilm activity, antioxidant activity

## Abstract

Developing new types of effective antimicrobial compounds derived from natural products is of interest for the food industry. Some analogs to A-type proanthocyanidins have shown promising antimicrobial and antibiofilm activities against foodborne bacteria. We report herein the synthesis of seven additional analogs with NO_2_ group at A-ring and their abilities for inhibiting the growth and the biofilm formation by twenty-one foodborne bacteria. Among them, analog **4** (one OH at B-ring; two OHs at D-ring) showed the highest antimicrobial activity. The best results with these new analogs were obtained in terms of their antibiofilm activities: analog **1** (two OHs at B-ring; one OH at D-ring) inhibited at least 75% of biofilm formation by six strains at all of the concentrations tested, analog **2** (two OHs at B-ring; two OHs at D-ring; one CH_3_ at C-ring) also showed antibiofilm activity on thirteen of the bacteria tested, and analog **5** (one OH at B-ring; one OH at D-ring) was able to disrupt preformed biofilms in eleven strains. The description of new and more active analogs of natural compounds and the elucidation of their structure-activity relationships may contribute to the active development of new food packaging for preventing biofilm formation and lengthening the food shelf life.

## 1. Introduction

The use of preservatives in the food industry has generated a high concern among consumers, due to the possible short, medium, and long-term health effects. As a result of the potential of synthetic preservatives to cause health problems, consumers and companies are trying to replace synthetic ones with natural preservatives, which can be achieved from sources such as plants, bacteria, fungi, animals, and algae and are considered safer for humans and the environment [1]. Consequently, there is a general search for innovation in the food industries in order to provide healthy and safe food, as well as a high interest in developing new types of effective antimicrobial compounds derived from natural sources.

Proanthocyanidins (PACs) are ubiquitous natural products that constitute one of the most important families of polyphenols in nature [2] and they are characterized by several biological activities, such as antidiabetic [3], anti-cancer [4], neuroprotective [5], however, are mainly antioxidant [6] and antimicrobial [7,8]. Due to their potent antioxidant and antimicrobial properties, PACs are also suitable for the preparation of active packaging films in the food industry [9], although this aspect has been rarely investigated.

Our research group has recently worked on the antimicrobial and antibiofilm activities of two natural PACs (cinnamtannin B-1 and procyanidin B-2) isolated from laurel (*Laurus nobilis* L.) wood extracts and six synthetic A-type PACs analogs (compounds **I**–**VI**) against several foodborne microorganisms [10,11] (Figure 1). Cinnamtannin B-1 (an A-type PAC) was found to have higher antimicrobial activity than procyanidin B-2 (a B-type PAC) [12] and for that reason several structurally-simplified analogs to cinnamtannin B-1 were designed (compounds **I**–**VI**), synthesized and evaluated for their antimicrobial and antibiofilm activities [11]. It was deduced from that study that the absence of electron-donating groups (OH groups) at A-ring increased the activity, as well as a smaller size of the bottom monomer. A relative higher polarity also improved the activity of the compounds. Among those analogs (Figure 1), compound **IV**, with a nitro group at A-ring, showed the highest antimicrobial activity in the set. Furthermore, it was one of the best compounds at preventing biofilm formation with more promising effects on the disruption of preformed biofilms. Thus, compound **IV** emerged as a new leading structure for further structure–activity studies (Figure 1).

With the purpose of obtaining additional analogs even more active than compound **IV** and studying the influence of the substitution pattern on rings B, C and D, we have now envisioned other seven analogs to A-type PACs, all of them with a nitro group at A-ring, as for compound **IV**, with one or two hydroxyl groups on rings B and D, and with a methyl group at C-ring or not (Figure 2). We therefore describe here the synthesis of analogs **1**–**7** following a procedure based on flavylium chemistry [12] and their antimicrobial and antibiofilm activities against both culture-type bacterial strains and foodborne bacteria from organic foods with high tolerance to biocides and resistance to antibiotics. We also conclude possible structure–activity relationships, in order to look for the most effective molecule to be used for the development of active packaging films based on PACs.

## 2. Results and Discussion

### 2.1. Synthesis of Analogs **1**–**7** and Their Antioxidant Activity

The synthetic route followed to prepare analogs **1**–**7** is outlined in Figure 1. These compounds have been synthesized by the nucleophilic attack of phloroglucinol (**17**) or resorcinol (**18**) on flavylium salts **13**–**16**, which were prepared through acid-catalyzed condensation of aldehyde **12** with acetophenone derivatives (**8**–**11**). The flavylium salts have been prepared following a classic method that uses a solution of sulfuric acid in acetic acid [13]. On the other hand, analogs **1**–**7** were synthesized following the general procedure B (see Section 3.3). According to our previous experience, this general procedure is the best method to achieve the nucleophilic addition between π-nucleophiles and flavylium salts with low electronic density [14]. Thus, all flavylium salts were able to react with **17** and **18** in methanol at 50 °C to give analogs **1**–**7** in moderate to good yields (45–83% from initial aldehyde) [15].

The structures of the synthesized compounds were confirmed by comparison of their ^1^H NMR and ^13^C NMR spectra with those reported in the literature [11,15,16].

Regarding the antioxidant activity of the synthesized analogs, compounds **1**–**3** showed (Table 1), as expected, a higher DPPH radical-scavenging activity than the rest because of the presence of the catechol moiety at B-ring. These compounds were around two-fold less active than the reference used (Trolox). It seems that the analog with phloroglucinol moiety (**2**) is slightly more antioxidant that those with resorcinol (**1** and **3**). Moreover, it also seems that the presence of a CH_3_ group slightly improved the ability of the analog for scavenging the DPPH radical (**1** vs. **3**).

### 2.2. Antimicrobial Activity

We had previously described the antimicrobial activity of compound **IV** [11], which showed MIC values of 10 μg/mL against *B. cereus* UJA27q and *S. saprophyticus* UJA27g and of 50 μg/mL against all the remaining strains analyzed except for *K. terrigena* UJA32j (MIC value of 100 μg/mL) and *Salmonella* sp. UJA40l (MIC of 1 mg/mL).

The standard agar diffusion method allowed us to develop a rough idea about antimicrobial potential of the new screened compounds, showing analog **4** the best results, with zones of inhibition of at least 10 mm when tested at a concentration of 1 mg/mL against the foodborne resistant strains *E. casseliflavus* UJA11e, *S. saprophyticus* UJA27g, *B. cereus* UJA27q, *P. agglomerans* UJA29o, *K. terrigena* UJA32j, *S. aureus* UJA34f and *L. casei* UJA35h, as well as 8 mm against *E. faecium* UJA11c and *Enterobacter* sp. UJA37p (Table 2). When 100 μg/mL was used as the initial concentration in these assays, diameters of inhibition of 12 mm were also achieved against *B. cereus* UJA27q and of 10 mm against *L. casei* UJA35h.

Values of minimal inhibitory concentrations corroborated analog **4** as the most active analog against mainly Gram positive target strains, showing MICs of 10 μg/mL against *S. saprophyticus* UJA27g, and of 50 μg/mL against *E. faecium* UJA11c, *E. casseliflavus* UJA11e, *B. cereus* UJA27q, *S. aureus* UJA34f, *L. casei* UJA35h, *P. agglomerans* UJA7m, and *Enterobacter* sp. UJA37p (Table 3a). Analogs **6** and **7** also showed a high antimicrobial activity against *S. saprophyticus* UJA27g, *B. cereus* UJA27q, *S. aureus* UJA34f and *L. casei* UJA35h, with MICs of 10 μg/mL against all of them.

The four Gram positive foodborne strains (*S. saprophyticus* UJA27g, *B. cereus* UJA 27q, *S. aureus* UJA34f and *L. casei* UJA35h) were particularly sensitive when incubated with most of the analogs, showing MICs of 10 and 50 μg/mL for almost all of them. Among culture type bacteria (Table 3b), *S. aureus* CECT828 was the most sensitive strain to the analogs **2**, **4** and **7**, showing MICs of 50 μg/mL, as well as *S. aureus* CECT976 as for analog **2**. MICs for all other type strains analyzed (*L. innocua* CECT 910, *E. coli* CCUG47553, *E. coli* CCUG47557, *S. enterica* CECT 4300, *S. enterica* CECT 409, *S. enterica* CECT 4395 and *S. enterica* CECT 915) were above 1 mg/mL for all of the analogs tested.

In order to look for possible synergistic combinations, the checkerboard titer test was applied to analog **4** together with all of the other compounds against *S. saprophyticus* UJA27g (Appendix A), *B. cereus* UJA27q (Appendix A) and *S. aureus* UJA34f (Appendix A), strains previously determined as particularly sensitive to these analogs. We have detected synergistic activities between analog **4** and analogs **2** and **5** against *S. saprophyticus* UJA27g, as well as between analog **4** and analogs **3**, **6** and **7** against *S. aureus* UJA34f. When analog **4** was combined with the other compounds, indifferent results (neither synergistic effects nor antagonisms) were obtained in the checkerboard assay against the three strains tested.

The best results with all of these new analogs were obtained on the inhibition of biofilm formation and the disruption of previously established biofilms by the target strains. Table 4 and Table 5 show the results of these assays, which reported many analogs at different concentrations being able to inhibit at least the 75% of the formation and/or disrupt at least the 75% of the established biofilms when compared to the control strains in culture media. These results are of great importance for food industries, as studies have shown that biofilm sanitizer tolerance is mainly correlated to biofilm mass development [17,18]. Mature biofilms are generally more tolerant to stressful conditions and antimicrobial treatments, due to the strong 3D structure established by the multiple layers of bacterial cells, which constitutes a strong physical barrier that limits and obstructs the penetration of sanitizers or biocides [19].

Analog **1** stands out by showing an inhibition of at least 75% of biofilm formation by the strains UJA7m, UJA11c, UJA11e, UJA27g, UJA27q and UJA29o at all of the concentrations tested, ranging from 10 µg/mL to 0.01 µg/mL, and it also inhibited the formation of biofilm by the other six strains mainly at low doses. Analog **2** also showed an inhibition of at least 75% of biofilm formation by thirteen of the bacteria tested, including culture type strains *S. aureus* CECT 828 and *S. aureus* CECT 976, and it also induced the disruption of preformed biofilms by twelve of the strains analyzed, including *S. aureus* CECT 976. Analog **7** had an inhibitory effect on biofilm formation by eleven strains, including *S. aureus* CECT 828 and it was also able to disrupt the biofilm previously formed by nine of the analyzed strains. High antibiofilm effects on Gram positive bacteria have also been described for the natural compound eugenol, which significantly suppresses adherence, the initial step in caries formation, by *Streptococcus mutans* compared with the control [20].

Analogs **3** and **4** inhibited the formation of biofilm by ten strains and analogs **5** and **6** had similar effects on nine of the bacteria tested. Disruption of preformed biofilms was achieved on eight to eleven strains by these four analogs, showing all of them to have similar results in their antibiofilm activities. The paradoxical effect detected in some of these analogs is remarkable, showing better activity at lower doses on the antibiofilm effects, as previously defined when studying cranberry proanthocyanidins and echinocandins [10,11,21]. As to the specific mechanisms of these anti-biofilm effects, changes in exopolysaccharide (EPS) production or motility in both Gram positive and Gram negative bacteria, as well as changes in hydrophobicity may account for the antibiofilm activities we have found in our analogs, as previously described for some natural and derived compounds [22]. However, further studies are necessary to corroborate this hypothesis.

The multiple antibacterial effects detected on foodborne bacteria are summarized in Figure 3, which shows key results of each of the studied analogs in both antimicrobial and antibiofilm activities, especially at very low concentrations.

The complex structure of biofilm provides them with enhanced resistance to stress, including cleaning and disinfection methods traditionally used in food processing plants. Therefore, it is urgent to find methods and strategies for effectively combating bacterial biofilm formation and eradicating mature biofilms [23]. As for the food industries, it has also been previously evidenced that proanthocyanidin-based chitosan films exhibit higher antioxidant and antimicrobial ability as compared with basic films, and the content of these compounds also has a great impact on the properties of these chitosan-based films [9], so the description of new and more active analogs of these natural compounds may contribute to the active development of new food packaging preventing biofilm formation by foodborne pathogens, and the consequent lengthening of food shelf life.

## 3. Materials and Methods

### 3.1. Chemicals and Instruments

Commercially available reagents were used without further purification. Phloroglucinol (**17**) (Sigma-Aldrich Chemie, Steinheim, Germany), resorcinol (**18**), aldehyde **12** and ketones **8**–**11** (Alfa Aesar, Thermo Fisher Scientific, Karlsruhe, Germany). All solvents used in the chemical syntheses and preparative chromatographies were commercially available and used as received (Panreac, AppliChem Gmbh, Darmstadt, Germany). Methanol used for high-performance liquid chromatography (HPLC) was of HPLC grade (VWR Chemicals, Prolabo, Fontenay-sous-Bois, France). Deuterated methanol (CD_3_OD) and acetonitrile (CD_3_CN) were used to prepare solutions of purified compounds for nuclear magnetic resonance (NMR). For flavylium salts, DCl was added to acidify the solution. Analytical thin-layer chromatography (TLC) was performed on silica gel 60 F_254_ precoated aluminum sheets (0.25 mm, Merck Chemicals, Darmsdadt, Germany). Silica gel 60, 200–400 mesh (Merck Chemicals, Darmsdadt, Germany), was used for silica gel column chromatography (CC), and Sephadex LH-20 (Sigma-Aldrich Chemie, Steinheim, Germany) for size-exclusion chromatography (SEC). Analytical HPLC analyses were performed on a C_18_ reversed-phase Spherisorb ODS-2 column, 250 mm × 3 mm i.d., 5 μm (Waters Chromatography Division, Milford, MA, USA). Semipreparative HPLC separations were performed on a C_18_ reversed-phase Spherisorb ODS-2 column, 250 mm × 10 mm i.d., 5 mm (Waters Chromatography Division, Milford, MA, USA) on the instrument described above, at flow rate of 5 mL/min. ^1^H NMR and ^13^C NMR spectra were recorded on a Bruker Avance 400 spectrometer (Bruker Daltonik GmbH, Bremen, Germany) operating at 400 and 100 MHz for ^1^H and ^13^C, respectively.

### 3.2. General Procedure A for the Synthesis of Flavylium Salts (**13**–**16**)

A mixture of aldehyde **12** (1 mmol), the acetophenone derivative (**8** or **9** or **10** or **11**, 1 mmol), 98% H_2_SO_4_ (0.3 mL; 5.4 mmol) and HOAc (1.3 mL) was stirred overnight at room temperature following a similar procedure to that described by Calogero et al. [13]. Then Et_2_O (30 mL) was added and a red solid precipitated. The solid was filtered off and carefully washed with Et_2_O and dried, yielding compounds **13** (77% yield) or **14** (85% yield) or **15** (76% yield) or **16** (91% yield), respectively. The structure of all these known starting flavylium salts was confirmed by comparison of their physical and spectral data (^1^H NMR and ^13^C NMR) with those reported in the literature [11,14,16,24].

### 3.3. General Procedure B for the Synthesis of 2,8-Dioxabicyclo[3.3.1]nonane (**1**–**7**)

A mixture of the flavylium salt derivative (**13** or **14** or **15** or **16**), and phloroglucinol (**17**) or resorcinol (**18**) (0.5 mmol) in absolute methanol (8 mL) was stirred overnight at 50 °C following a similar procedure to that described by Kraus et al. [15]. Then, the solvent was removed and the crude was purified by semipreparative HPLC, silica gel column chromatography (CC) or size-exclusion chromatography (SEC) to give analogs **1** (50% from **12**) or **2** (83% from **12**) or **3** (60% from **12**) or **4** (45% from **12**) or **5** (64% from **12**) or **6** (50% from **12**) or **7** (64% from **12**), respectively. The structure of all these known dioxabicyclo[3.3.1]nonane derivatives was confirmed by comparison of their physical and spectral data (^1^H NMR and ^13^C NMR) with those reported in the literature [11,15,25].

### 3.4. DPPH Radical-Scavenging Activity

The radical-scavenging activity of analogs **1**–**7** and Trolox (reference antioxidant) against the stable DPPH radical was determined spectrophotometrically in a Genesys^TM^ 150 Vis/UV–Vis spectrophotometer (Thermo Fischer Scientific, Waltham, MA, USA), following a modified procedure based on the literature and reported by the authors [16]. Methanolic solutions (2.4 mL) of DPPH (4.7 × 10^−5^ M) with an absorbance at 515 nm of 0.800 ± 0.030 AU were mixed with methanolic solutions (1.2 mL) of samples at different concentration (1–1000 ppm) by dissolving dry samples in methanol. The experiment was carried out in triplicate. The samples were shaken and allowed to stand for 15 min. in the dark at room temperature and then the decrease in absorbance was measured at 515 nm. The radical-scavenging activity was expressed in terms of the antioxidant concentration (µM) required to decrease the initial DPPH^•^ concentration by 50% (Effective Concentration: EC_50_). The percentage of the DPPH^•^ remaining, calculated by the following equation:% DPPH rem = [DPPH]/[DPPH]_0_ × 100
where [DPPH] is the concentration of DPPH^•^ at the time measured (t = 15 min) and [DPPH]_0_ is the initial concentration of DPPH^•^ (t = 0 min), was plotted against the sample concentration (µg/mL), a linear or logarithmic regression curve being established in order to calculate the EC_50_ (Table 1).

### 3.5. Antimicrobial Activity

We firstly screened the antimicrobial activity of the analogs by using the standard agar diffusion method. In terms of the results obtained, we determined the minimal inhibitory concentration (MIC) values for each sample. As targets for these assays, we have used strains from the Spanish Type Culture Collection (CECT), the Culture Collection of the University of Goteborg (CCUG), as well as strains from our own collection from organic foods, showing high tolerance to biocides and resistance to antibiotics [26]. Bacterial strains are listed in Table 6. All experiments were carried out in triplicate.

### 3.6. Minimal Inhibitory Concentration (MIC) Test

The optimal concentrations of each compound to be used in MIC tests was derived from results of standard agar diffusion tests. MIC values were determined by the broth microdilution method as recommended by the Clinical and Laboratory Standards Institute [27].

### 3.7. Checkerboard Titer Tests

The possible synergistic effects between the most active analogs and all the other compounds were evaluated by the checkerboard method and expressed as the sum of the fractional inhibitory concentration (FIC) index for each agent, calculated as the MIC of this agent in combination divided by the MIC of this agent alone. The FIC value of the most effective combination is used in calculating the fractional inhibitory concentration index (FICI) by adding both FICs: FICI = FICA + FICB = CAcomb/MICAalone + CBcomb/MICBalone, where MICAalone and MICBalone are the MICs of drugs A and B when acting alone and CAcomb and CBcomb are concentrations of drugs A and B at the isoeffective combinations, respectively. The FICI was interpreted as synergistic when it was ≤0.5, antagonistic when it was >4.0, and any value in between was interpreted as indifferent [28,29]. Each isolate was tested in triplicate.

### 3.8. Biofilm Formation Inhibition Assay

The capacity of the compounds in obstructing biofilm formation was determined by incubating target strains with 10-fold serially diluted purified compounds, ranging from 0.1 μg/mL to 10 μg/mL, based on the MIC values previously obtained, as described by Ulrey et al. [30]. Inhibition of biofilm formation induced by isolated compounds was measured by the crystal violet stain method as previously described by us [11].

### 3.9. Disruption of Preformed Biofilm

Cells were allowed to settle biofilms during 24 h, previously to the addition of appropriate diluted compounds, and after a second incubation (24 h, 30 °C) remaining biofilm was measured by the crystal violet stain method, as described for the biofilm formation inhibition assay.

### 3.10. Statistical Analysis

The average data and standard deviations from absorbances were determined with Excel program (Microsoft Corp., Redmond, WA, USA). A *t*-test was performed at the 95% confidence level with Statgraphics Plus version 5.1 (Statistical Graphics Corp., Rockville, MD, USA), to determine the statistical significance of data.

## 4. Conclusions

In this work, seven analogs to the natural A-type proanthocyanidins have been synthesized and their antimicrobial and antibiofilm activities have been established. All of these compounds were designed with an electron-withdrawing group (NO_2_) on the A-ring (at carbon **6**), since the most active compound found in a previous work had that structural feature, and the differences among them are in the number of OH groups on rings B and D (one or two) and in the presence or absence of a methyl group at C-ring. Regarding the antimicrobial activity, it seems that (a) the analogs with only one OH group at B-ring (**4**, **7**, **6**) are more active than those with two OH groups (**1**, **3**, **2**) (just the opposite of what happens with antioxidant activity), (b) the analogs with three (**4**, **6**) or two (**7**) OH groups in total are more active than that with four OH groups (**2**), and (c) the presence (**7**, **6**) or absence (**4**, **1**) of a methyl group at C-ring is not determinative for activity. Taking into consideration both the inhibition of biofilm formation and the disruption of preformed biofilms, analog **2** (with two OH groups at B-ring, two OH groups at D-ring and a methyl group at C-ring) is the most effective, especially at low concentrations. Furthermore, it is the most active analog on culture type strains *Staphylococcus aureus* CECT 976 and CECT 828 in terms of antimicrobial activity. On the other hand, analog **2** also showed the highest antioxidant activity. Other compounds with good antibiofilm activities were **1**, **5**, **4** and **7**, without being able to establish common structural features for all of them.

## Data Availability

Not applicable.

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
