# Peer review of "Synthesis of Analogs to A-Type Proanthocyanidin Natural Products with Enhanced Antimicrobial Properties against Foodborne Microorganisms"

_molecules, 2023, doi:10.3390/molecules28124844_

Round 1
Reviewer 1 Report
As presented in the manuscript, Scheme 1 is virtually unreadable. Mainly it seems to have been distorted in the copying process, although it's possible that too much information has been crammed into the scheme. Conversely, Figures 1 and 2 have used a lot of space to present structures that could be condensed. The information about the assays needs to be presented in a clearer format so that the key results are more evident. The research idea and the results seem reasonable enough.
The manuscript as submitted needs editing for a variety of minor English problems, which I would have helped with had there not been other flaws in the presentation of the science. If and when you redo the manuscript, I am willing to do additional editing, as I usually do.
Author Response
Point 1: As presented in the manuscript, Scheme 1 is virtually unreadable. Mainly it seems to have been distorted in the copying process, although it's possible that too much information has been crammed into the scheme.
Response 1: Regarding Scheme 1, it was likely distorted in the copying process, since it was pasted directly from the ChemDraw program into the template. It has now been replaced by a Tif image to prevent some letters and numbers from moving with respect to the structure drawings.
Point 2: Conversely, Figures 1 and 2 have used a lot of space to present structures that could be condensed.
Response 2: Figure 1 has been condensed, drawing a common structure for compounds II-V. Thank you, it has been a good idea. However, we prefer to leave Figure 2 as it is, as readers may better appreciate the structural differences among analogues (with separate drawings for each molecule) and see the structure-activity relationship more easily.
Point 3: The information about the assays needs to be presented in a clearer format so that the key results are more evident. The research idea and the results seem reasonable enough.
Response 3: In order to present the results of the antimicrobial assays in a clearer format, we have included in the manuscript a Figure (numbered 3), which shows the key results of each of the studied analogues in both antimicrobial and antibiofilm activities, especially at very low concentrations. It presents in a graphical and immediate view the key results of each of the compounds tested.
Reviewer 2 Report
The present manuscript by Antonio Cobo et al describes the preparation of seven nitro analogues to A-type proanthocyanidins and the study of their antioxidant, antimicrobial and inhibitory to biofilm formation activities. The results were promising and certain SAR points were suggested which could be of interest to researchers in the field. However, it is not clear in the manuscript which of the reported compounds (intermediates or final products) are new. Thus, if the compounds are new the authors should report spectral, mps (for solid compounds) and purity data; otherwise, every known compound should be specifically/appropriately referred and no experimental procedure is necessary. Minor revision: line 87, “(see section 3)” should change to “(see section 4.3).
Minor editing of English language required
Author Response
Point 1: It is not clear in the manuscript which of the reported compounds (intermediates or final products) are new. Thus, if the compounds are new the authors should report spectral, mps (for solid compounds) and purity data; otherwise, every known compound should be specifically/appropriately referred and no experimental procedure is necessary.
Response 1: All flavylium salts (intermediates) and analogues (final products) are already known, so most of the text has been omitted from (new) sections 3.2 and 3.3 and the corresponding references have been added, as recommended.
Point 2: Minor revision: line 87, “(see section 3)” should change to “(see section 4.3).
Response 2: Material and methods was wrongly numbered as section 4 in the original manuscript. It has been amended in the new version, an now it is numbered as section 3, so it has been changed to "see section 3.3" (line 87).
Reviewer 3 Report
Natural and synthetic antioxidants are widely used in the food industry. But their effect on the microflora of the gastrointestinal tract requires constant study, which can reveal negative side effects. Synthetic derivatives make it possible to increase the beneficial properties of such food additives and reduce the negative consequences of their use, unlike traditionally used natural antioxidants. The revealed regularities about the influence of certain structural elements will make it possible to predict more effective and safe options.
The manuscript is quite well designed, the experimental part is described in detail in the general part. The conclusions are based on the results obtained.
The manuscript may be of interest to synthetic chemists as well as food technologists. It may be accepted for publication after minor corrections.
Clarification needed
Line 142
a MIC was above 1mg/mL
The description of the symbol b is missing
Author Response
Point 1: Line 142: a MIC was above 1mg/mL. The description of the symbol b is missing
Response 1: Symbol b was a mistake in the previous version of the manuscript. The only symbol included in table 3a is a, which is adequately described.
Round 2
Reviewer 1 Report
The modifications in Figure 1 work very well.
In my copy of the MS there is an issue in Scheme 1, likely from the copying procedure from ChemDraw again, where the Scheme has been duplicated.
In my copy of Table 3a in row 2 it looks like there are errors in the first 2 entries.
Aside from some formatting problems the manuscript reads well.
A few minor editorial suggestions follow:
l. 37 Delete "request"
l. 44 Delete"to be used"
l. 97 Does the "a" after (18) mean anything?
l. 145 Change "These" to "The"
l. 149 Delete "when faced"
l. 363 EC50 is "Effective Concentration"
Author Response
Point 1: In my copy of the MS there is an issue in Scheme 1, likely from the copying procedure from ChemDraw again, where the Scheme has been duplicated.
Response 1: Scheme 1 has been revised and in the new version it is not duplicated, maybe it seemed like that because of the use of the"Track Changes” function in MS Word.
Point 2: In my copy of Table 3a in row 2 it looks like there are errors in the first 2 entries.
Response 2: It is probably also due to the "Track Changes” function we have been using in MS Word. It has also been revised in the new version.
Point 3: Few minor editorial suggestions
Response 3: All editorial suggestions have been amended as recommended.
Regarding the "a" after (18) (line97), it refers to the symbol in the box of the scheme 1, realtive to the reagents and conditions used.
Reviewer 2 Report
In the present version of their manuscript the authors, Antonio Cobo et al, made all the suggested revisions. Thus, I recommend its publication.
Author Response
Thank you very much for your previous suggestions and the recommendation for publication.
Best regards